# Cost-Effectiveness of Dexamethasone Intravitreal Implant in Naïve and Previously Treated Patients with Diabetic Macular Edema

**DOI:** 10.3390/ijerph20085462

**Published:** 2023-04-11

**Authors:** Marta Medina-Baena, Olga Cejudo-Corbalán, Fernando Labella-Quesada, Eloy Girela-López

**Affiliations:** 1Puerta del Mar University Hospital, 11009 Cadiz, Spain; 2Puerto Real University Hospital, 11510 Cadiz, Spain; 3Faculty of Medicine and Nursing, University of Córdoba, 14004 Cordoba, Spain

**Keywords:** diabetic macular edema, treatment naïve, dexamethasone intravitreal implant, VEGF inhibitors, cost-effectiveness ratio, incremental cost-effectiveness ratio

## Abstract

Purpose: To compare the direct costs associated with the dexamethasone intravitreal implant (DEX-i) in treatment-naïve and previously treated eyes with diabetic macular edema (DME) in a real clinical setting. Methods: Retrospective and single-center study conducted in a real clinical scenario. Consecutive DME patients, either naïve or previously treated with vascular endothelial growth factor inhibitors (anti-VEGF), who received treatment with one or more DEX-i between May 2015 and December 2020, and who were followed-up for a minimum of 12 months, were included in the study. The cost analysis was performed from the perspective of the Andalusian Regional Healthcare Service. The primary effectiveness endpoint was the probability of achieving an improvement in best-corrected visual acuity (BCVA) ≥ 15 ETDRS letters after 1 year of treatment. The incremental cost-effectiveness ratio (ICER) of different improvements in BCVA was calculated. Results: Forty-nine eyes, twenty-eight (57.1%) eyes from the treatment-naïve group and twenty-one (42.9%) from the previously treated group, were included in the analysis. The total cost of one year of treatment was significantly lower in the treatment-naïve eyes than in the previously treated eyes [Hodges-Lehmann median difference: EUR 819.1; 95% confidence interval (CI): EUR 786.9 to EUR 1572.8; *p* < 0.0001]. The probability of achieving a BCVA improvement of ≥15 letters at month 12 was significantly greater in the treatment-naïve group than in the previously treated group (rate difference: 0.321; 95% CI: 0.066 to 0.709; *p* = 0.0272). The Cochran–Mantel–Haenszel Odds Ratio of achieving a BCVA improvement of ≥15 letters at month 12 was 3.55 (95% CI: 1.09 to 11.58; *p* = 0.0309). In terms of ICER, the treatment-naïve group showed cost savings of EUR 7704.2 and EUR 5994.2 for achieving an improvement in BCVA ≥ 15 letters at month 12 and at any of the measured time points, respectively. Conclusions: DEX-i was found to be more cost-effective in treatment-naïve eyes than in those previously treated with anti-VEGF. Further studies are needed to determine the most cost-effective treatment based on patient profile.

## 1. Introduction

Recent years have seen major socio-demographic changes that have inevitably been challenges to health systems. While these changes started initially in high-income countries, it is now low- and middle-income countries that are experiencing these problems. It is estimated that by 2050, approximately 1 in 6 people in the world will be over the age of 65, compared to 1 in 11 in 2019 [1].

As the world’s elderly population continues to grow and age, healthcare costs are expected to rise further. Healthcare systems must therefore be efficient to meet the growing demand of cases affected by chronic diseases [2].

The prevalence of diabetes mellitus (DM), particularly type 2 DM, has increased rapidly in recent decades [3,4]. It is estimated that approximately 693 million people will have DM by 2045 [5], with 66.7 million people having DM in the region of Europe alone [6].

Although in recent decades we have seen a decrease in the incidence of diabetic retinopathy (DR) [7], approximately 50% of patients fail to achieve good control of their DM [8]; thus, the risk of complications remains significant.

Diabetic retinopathy is the most common cause of blindness in the working-age population (approximately 2.6% of all vision loss), and diabetic macular edema (DME) secondary to DR is the direct cause of visual impairment [9,10,11].

The pathogenesis of DME is multifactorial and complex. Both vasogenic mediators, such as vascular endothelial growth factors (VEGF), and inflammatory mediators, such as inflammatory cytokines, are involved [12].

Although VEGF inhibitors (anti-VEGF) are commonly used as first-line treatment for DME [13,14], up to 30–50% of patients do not respond adequately to anti-VEGF therapy [15,16].

Intravitreal dexamethasone implant (DEX-i) has been shown to positively impact both anatomical and functional clinical outcomes, as it acts on inflammatory and angiogenic mediators [17,18,19]. In addition, patients who do not respond adequately to anti-VEGF therapy should be switched to DEX-i as soon as possible [20,21].

Due to the high cost of DME treatment, the appropriateness of new therapies must be based on patient benefit.

Despite the clinical, social and economic relevance of this therapeutic area, data evaluating the cost effectiveness of DEX-i in DME are limited [22,23,24,25].

The aim of this study was to compare the direct costs associated with DEX-i in treatment-naïve and previously treated eyes with DME in a real clinical setting.

To the authors’ knowledge, this is the first study to evaluate the budgetary impact of this therapy from the perspective of the Andalusian Regional Healthcare Service (ARHS), and may therefore favor optimization of public financial resources and provide patient-tailored solutions useful for decision-making processes.

## 2. Materials and Methods

### 2.1. Study Design

The study was a retrospective single-center study conducted in a real clinical setting.

The study protocol was approved by the ethics committee of the University Hospital of Puerto Real, approval number (MMB-NAIVE-2018), which waived the need for written informed consent to participate in this study. However, all participants gave written informed consent prior to DEX-i injection.

This study is in accordance with the principles of the Declaration of Helsinki and the Good Clinical Practice/International Council for Harmonization Guidelines.

### 2.2. Participants

Consecutive DME patients, either treatment-naïve or previously treated with anti-VEGF agents, treated with one or more injections of the Ozurdex^®^ implant between May 2015 and December 2020 and followed up for at least 12 months.

### 2.3. Inclusion/Exclusion Criteria

This study included treatment-naïve and previously treated DME patients aged ≥ 18 years; baseline BCVA ≥ 5 letters (ETDRS charts); glycosylated hemoglobin A1c (HbA1c) ≤ 10%; and who had at least a minimum DEX implant follow-up period of 12 months after DEX implant.

Patients with macular edema secondary to any other condition; with presence of vitreomacular diseases, including, but not limited to, macular ischemia, vitreomacular traction, foveal atrophy, etc.; with intraocular pressure (IOP) ≥ 25 mmHg; and with history of vitrectomy were excluded.

### 2.4. Direct Costs

Cost analysis was carried out from the ARHS perspective.

Healthcare system costs were obtained from the ARHS. The cost of medical supplies and consumables, as well as the cost of antiangiogenic treatments, were calculated according to the information provided by the study center.

Costs are expressed in euros (EUR) and have been updated for the year 2020. A summary of the unit costs is shown in Annex I.

Total direct healthcare costs and average costs per patient were calculated. The total cost was estimated considering the unit cost of different resources and the amount of resource consumed per patient.

### 2.5. Definitions

DME was defined as a retinal thickening (≥250 µm) within one disk diameter of the center of the macula [26].

A treatment-naïve patient was defined as a patient who, up to the time of recruitment for the study, had never received any pharmacological, laser and/or surgical treatment [27].

DME was considered to be anatomically resolved when the central retinal thickness (CRT) was <250 µm and there were no signs of edema in the OCT.

### 2.6. Calculations

Cost-effectiveness ratio (CER) was calculated according to the cost/result formula. The CER was calculated once, taking into account the numerical difference in outcomes.

The incremental cost-effectiveness ratio (ICER) was calculated using the numerical difference in outcome between the untreated and treated groups using the following formula: ICER = CER for the untreated group—CER for the treated group.

### 2.7. Outcomes

The effect of DEX-i was estimated in terms of best-corrected visual acuity (BCVA) gain.

The primary effectiveness endpoint was the probability of achieving an improvement in BCVA ≥ 15 ETDRS letters after 1 year of treatment. The ICER of ≥15 letters gained for DEX-i between treatment-naïve and previously treated patients was calculated as the final outcome of the model.

ICER was also calculated for ≥15 ETDRS letters gained at some point in follow-up; ≥10 letters gained at some point in follow-up; and ≥10 letters gained after 1 year of treatment.

### 2.8. Statistical Analysis

Standard statistical analyses were performed with the statistical program MedCalc^®^ statistical software version 20.104 (MedCalc Software Ltd., Ostend, Belgium; https://www.medcalc.org; accessed on 23 November 2022).

Descriptive statistics number (percentage), mean [standard deviation (SD)], mean [95% confidence interval (95% CI)] or median [Interquartile range (IqR)] were used as appropriate.

The distribution of quantitative variables was assessed using the D’Agostino–Pearson test.

The Mann–Whitney U test was used to assess baseline quantitative variables between naive eyes and previously treated eyes.

The Cochran–Mantel–Haenszel test was used to assess the probability of achieving a BCVA improvement ≥ 15 letters (at any measured time point and at month 12) or ≥10 letters (at any measured time point and at month 12) with study group as the grouping variable and lens status as the factor variable.

Analysis of covariance (ANCOVA) was used to assess variation in BCVA and CRT between study groups. The model included “Group” (treatment-naïve versus previously treated) as a factor and age, DME duration and lens status as covariates.

## 3. Results

Forty-nine eyes were included in the analysis: twenty-eight (57.1%) eyes in the treatment-naïve group and twenty-one (42.9%) in the previously treated group.

Baseline demographic and clinical characteristics are summarized in Table 1.

The total cost of one year of treatment was significantly lower in the treatment-naïve eyes (median: EUR 2678.3; interquartile range: EUR 2678.3 to EUR 2678.3) than in the previously treated eyes (median: EUR 3465.2; interquartile range: EUR 3465.2 to EUR 4251.1) (Hodges–Lehmann median difference: EUR 819.1; 95% confidence interval: EUR 786.9 to EUR 1572.8; *p* < 0.0001).

A summary of the study costs is shown in Table 2.

The probability of achieving an improvement of BCVA ≥ 15 letters at month 12 was significantly higher in the treatment-naïve group (probability: 0.607; 95% CI: 0.354 to 0.972) than in the previously treated group (probability: 0.286; 95% CI: 0.105 to 0.622) (rate difference: 0.321; 9% 5 CI: 0.066 to 0.709; *p* = 0.0272).

The Cochran–Mantel–Haenszel Odds Ratio of achieving a BCVA improvement ≥ 15 letters at month 12 was 3.55 (95% CI: 1.09 to 11.58; *p* = 0.0309) (Table 3).

The CER for an improvement of ≥15 letters in BCVA at 12 months was EUR 4411.9 in the treatment-naïve group and EUR 12,116.1 in the previously treated group. All measurements in the treatment-naïve group showed lower CERs (Table 4).

Regarding ICER, the treatment-naïve group achieved cost savings of EUR 7704.2 and EUR 5994.2 at month 12 and ≥15 letters in BCVA improvement at each measured time point, respectively (Table 4).

The median number of ETDRS letters gained after treatment with the DEX-i was significantly higher in the treatment-naïve group than in the previously treated group at months 2, 4, 8 and 12 (Figure 1A). However, BCVA was significantly improved at all the time points measured in both groups (*p* < 0.001 in each, as compared to baseline values. Friedman test).

The reduction in CRT was not significantly different between groups throughout the study, although CRT was significantly reduced compared to baseline at all time points measured (each *p* < 0.001 versus baseline. Friedman test) (Figure 1B).

After adjusting by covariates, the mean improvement in BCVA was significantly greater in the treatment-naïve group than in the previously treated group at months 2, 8 and 12 (*p* = 0.0134; *p* = 0.0060; and *p* = 0.0092, respectively) (Table 5). With the exception of month 6 (*p* = 0.0463), there were no significant differences in CRT reduction between treatment-naïve and previously treated eyes (Table 5).

The median number (IqR) of DEX-i administered throughout the study was 2.0 (1.5 to 2.0) in the treatment-naïve and 2.0 (2.0 to 3.0) in the previously treated group (Hodges–Lehmann median difference: 0.0; 95% CI: 0.0 to 1.0; *p* = 0.0124).

Regarding the cost analysis in patients with subretinal fluid, in the comparison of the cost adjusted by group and SRF, the presence of SRF was not relevant (*p* = 0.531), nor the interaction between both variables (*p* = 0.825).

## 4. Discussion

DME and DR have a significant economic impact on health systems, not only because of direct costs, but also because of indirect costs, such as reduced income or an increased need for social support as vision worsens [28]. Furthermore, it is estimated that the total cost per patient with DME is 30% higher than that of patients with DM without DME [29].

Although DME is a vision-threatening disease, it is a pathology with several treatment options [13,15]. Furthermore, the choice of treatment regimen for DME depends on the individual clinical characteristics of the patient [13].

Because healthcare systems have to face practically unlimited demand with limited resources, it is extremely important to identify cost-effective treatments.

In the present study, we compared the direct costs of DEX-i in treatment-naïve patients and those with an inadequate response to anti-VEGF.

According to the results of this study, when administered to treatment-naïve eyes, DEX-i was associated with a cost saving of EUR 7704.2 and EUR 5994.2 for achieving a BCVA improvement ≥ 15 letters at month 12 and at any of the time points measured, respectively. Furthermore, the CER of BCVA improvement ≥ 15 letters at month 12 was 2.75 times lower in the treatment-naïve group than in the previously treated group.

The economic impact of the introduction of DEX-i for the treatment of DME on the Spanish National Health System was estimated using a 3-year budget impact model [30]. The inclusion of intravitreal dexamethasone implant would result in an annual cost savings of (EUR, 2016) 35,030 during the first year of treatment [30]. After applying inflation (according to data from the Spanish Consumer Price Index as of 31 December 2021), the cost savings during the first year would be EUR 39,612 [30]. This figure is much higher than that observed in our study, where the total cost savings of the DEX-i (without taking into account the achievement of targets) was EUR 819.1. However, it is essential to note that the Cervera et al. study [30] was an estimate, while ours reflects a real-life analysis. Although both studies were conducted in Spain, it is not easy to compare our results with those of Cervera et al. [30]. While Cervera et al. compared the cost savings that would result from the introduction of DEX-i in the Spanish market, our study analyzed the difference in costs between patients who received DEX-i as their first therapy and those who received DEX-i after a failed anti-VEGF therapy.

In our study, each BCVA letter gained in the treatment-naïve group represented a saving of EUR 540.0 compared to the previously treated group.

In fact, all direct costs, with the exception of the first consultation and the cost associated with cataract surgery, were significantly lower in the treatment-naïve group than in the previously treated group.

Treatment-naïve eyes were 3.55 times more likely to have an improvement of ≥15 letters in BCVA at 12 months compared to previously treated eyes. This finding supports the hypothesis that naïve eyes benefit more from the applied treatment.

Several studies have reported the effectiveness of DEX-i implants in eyes with an inadequate VEGF response. Bush et al., in a real-life study, observed that, in eyes with an inadequate anti-VEGF response, switching patients to DEX-i produced better functional and anatomical outcomes than maintaining anti-VEGF treatment [20]. Similarly, Ruiz-Medrano et al. observed that, in eyes who did not adequately respond to anti-VEGF therapy after three injections, switching to DEX provided better functional outcomes [21].

Multiple hypotheses relate certain anatomical characteristics with a better response to DEX-I implants, such as the presence of subretinal fluid. In a previous analysis by the authors [31], the presence of subretinal fluid was significantly associated with the proportion of patients achieving a BCVA improvement ≥5 letters. This could lead to the conclusion that the cost of treatment in these patients would be reduced if DEX-i implant was used as a first-line treatment. However, in our study, in the comparison of the cost adjusted by group and SRF, the presence of SRF was not relevant (*p* = 0.531), nor the interaction between both variables (*p* = 0.825).

On the other hand, the design of this study did not have the objective of analyzing these anatomical characteristics or the cost in relation to them.

After adjusting for different covariates, visual acuity improvement was significantly greater in treatment-naïve eyes than in the previously treated eyes at months 2, 8 and 12. These findings are consistent with the results of previously published studies that indicated a greater improvement in BCVA in naïve compared with non-naïve DME eyes [32,33,34,35]. In our study, as in that published by Iglicki et al. [33], there was no significant difference at baseline visual acuity between the naïve and previously treated eyes.

Regarding CRT, there was a significant reduction at month 12 in the difference from baseline in naive and previously treated eyes (*p* < 0.0001, each, respectively), which is consistent with previous published studies [32,33,34,35].

This study has a number of limitations that need to be taken into account. In our opinion, the first and most important is its retrospective design. Bias and potential pitfalls are inherent to retrospective studies. The second limitation is the single-center nature of the study, with a limited number of patients. In addition, costs associated with potential adverse effects beyond cataract surgery were not taken into account. Finally, our study did not evaluate direct non-medical costs (i.e., home healthcare and social services), patient transport and other ancillary costs to determine economic parameters.

## 5. Conclusions

DEX-i proved to be more cost-effective in untreated eyes than in eyes previously treated with anti-VEGF. The costs of untreated patients were significantly lower in all aspects except for the costs associated with cataract surgery. In conclusion, this study demonstrates that DEX-i is effective in treating eyes with DME even in difficult cases where previous anti-VEGF treatment has failed, although functional outcomes were better in untreated eyes. However, this study did not compare the cost-effectiveness of DEX-i versus anti-VEGF in treatment-naïve eyes.

VEGF inhibitors are currently the first-line treatment for DME patients. However, many patients respond inadequately. Further studies, mainly prospective and multicenter, are therefore warranted to determine the most cost-effective treatment according to the patient profile.

## Figures and Tables

**Figure 1 ijerph-20-05462-f001:**
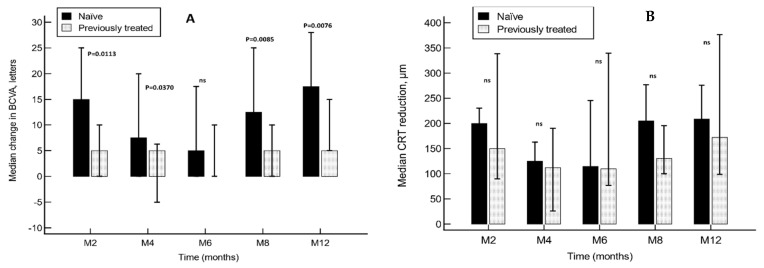
Median change in best-corrected visual acuity (**A**) and central retinal thickness (**B**) in treatment-naïve and previously treated eyes. Vertical bars represent interquartile range. CRT reduction is shown in absolute value. ns: not significant; BCVA: best-corrected visual acuity; CRT: central retinal thickness.

**Table 1 ijerph-20-05462-t001:** Main demographic and clinical characteristics of the study sample.

Variable	Overall (n = 49)	Naïve (n = 28)	Previously Treated (n = 21)	*p*
Age, years				0.0056 ^a^
Mean (SD)	67.7 (11.0)	64.4 (10.8)	72.1 (9.8)	
95% CI	64.6 to 70.9	60.2 to 68.6	67.6 to 76.6	
Median (IqR)	70.0 (61.9 to 75.9)	66.1 (57.0 to 70.6)	73.9 (67.4 to 80.6)	
Sex, n (%)				0.7736 ^b^
Men	23 (46.9)	14 (50.0)	9 (42.9)	
Women	26 (53.1)	14 (50.0)	12 (57.1)	
HbA1c, %				0.0992 ^a^
Mean (SD)	8.0 (1.3)	8.3 (1.4)	7.7 (1.1)	
95% CI	7.7 to 8.4	7.8 to 8.9	7.2 to 8.2	
Median (IqR)	7.9 (6.8 to 9.0)	8.4 (7.4 to 9.1)	7.1 (6.8 to 8.7)	
Lens status, n (%)				0.6831 ^b^
Phaquic	25 (51.0)	15 (53.6)	10 (47.6)	
Pseudophaquic	24 (49.0)	13 (46.4)	11 (52.49)	
Length of diabetes, years				0.8623 ^a^
Mean (SD)	13.8 (5.5)	14.1 (6.4)	13.4 (4.3)	
95% CI	12.2 to 15.4	11.6 to 16.6	11.5 to 15.4	
Median (IqR)	14.0 (10.0 to 17.8)	13.0 (10.0 to 20.0)	15.0 (10.0 to 15.3)	
Length of DME, months				<0.0001 ^a^
Mean (SD)	5.9 (3.9)	4.8 (4.6)	7.3 (1.9)	
95% CI	4.7 to 7.0	3.0 to 6.5	6.5 to 8.2	
Median (IqR)	6.0 (3.0 to 7.0)	3.5 (2.0 to 6.0)	6.0 (6.0 to 9.0)	
Previous treatment, n (%)		N.A.		N.A.
Bevacizumab	3 (14.3)		3 (14.3)	
Ranibizumab	6 (28.6)		6 (28.6)	
Aflibercept	12 (57.1)		12 (57.1)	
HRD, n (%)				0.0471 ^b^
Yes	37 (75.5)	18 (64.3)	19 (90.5)	
No	12 (24.5)	10 (35.7)	2 (9.5)	
SRF, n (%)				0.0696 ^b^
Yes	17 (34.7)	13 (46.4)	4 (19.0)	
No	32 (65.3)	15 (53.6)	17 (81.0)	
DRILS, n (%)				0.5637 ^b^
Yes	28 (57.1)	17 (60.7)	11 (52.4)	
No	21 (42.9)	11 (39.39	10 (47.6)	
IOP, mm Hg				0.0748 ^a^
Mean (SD)	15.6 (2.9)	15.0 (2.9)	16.4 (2.6)	
95% CI	14.8 to 16.4	13.9 to 16.1	15.2 to 17.6	
Median (IqR)	15.0 (14.0 to 18.0)	14.0 (12.5 to 17.5)	16.0 (15.0 to 18.0)	
BCVA, letters *				0.2157 ^a^
Mean (SD)	44.3 (22.5)	47.3 (24.1)	40.2 (20.0)	
95% CI	37.8 to 50.8	38.0 to 56.7	31.1 to 49.4	
Median (IqR)	45.0 (30.0 to 60.0)	47.5 (32.5 to 65.0)	35.0 (27.5 to 56.3)	
BCVA Pre-VEGF, letters *		N.A.		N.A.
Mean (SD)	44.5 (19.3)		44.5 (19.3)	
95% CI	35.7 to 53.3		35.7 to 53.3	
Median (IqR)	40.0 (30.0 to 60.0)		40.0 (30.0 to 60.0)	
CRT, µm				0.6936 ^a^
Mean (SD)	485.0 (129.3)	487.6 (111.2)	481.6 (153.0)	
95% CI	447.9 to 522.2	444.5 to 530.7	411.9 to 551.2	
Median (IqR)	478.0 (400.0 to 550.0)	478.5 (410.0 to 535.0)	450.0 (366.0 to 596.3)	
CRT Pre-VEGF, µm		N.A.		N.A.
Mean (SD)	452.6 (122.3)		452.6 (122.3)	
95% CI	382.0 to 523.3		382.0 to 523.3	
Median (IqR)	444.0 (380.0 to 520.0)		444.0 (380.0 to 520.0)	

^a^ Mann–Whitney U test (between naïve and non-naïve patients). ^b^ Fisher’s exact test. * Letters in the Early Treatment Diabetic Retinopathy Study (ETDRS) charts. SD = Standard deviation; CI = Confidence interval; IqR = interquartile range; DME = diabetic macular edema; HRD = high-reflective dots; SRF = subretinal fluid; DRILS = internal limiting membrane disruption; BCVA = best-corrected visual acuity; Pre-VEGF: before treatment with vascular endothelial growth factor inhibitors; CMT = central macular thickness; NA = not applicable.

**Table 2 ijerph-20-05462-t002:** Comparison of the different costs considered in the study between treatment-naïve and previously treated groups. *p* value was calculated using the Mann–Whitney test.

Costs	Treatment Naïve	Previously Treated	*p*
First medical visit			1.000
Mean (SD)	114.1 (0.0)	114.1 (0.0)	
Median (IqR)	114.1 (114.1 to 114.1)	114.1 (114.1 to 114.1)	
Medical visits *			<0.0001
Mean (SD)	270.9 (10.3)	327.5 (0.0)	
Median (IqR)	272.9 (272.9 to 272.9)	327.5 to 327.5)	
OCT			<0.0001
Mean (SD)	715.1 (22.7)	839.3 (0.0)	
Median (IqR)	719.4 (719.4 to 719.4)	839.3 (839.3 to 839.3)	
Cataract surgery			0.4185
Mean (SD)	217.8 (799.9)	435.7 (1093.6)	
Median (IqR)	N.A.	N.A.	
Anti-VEGF	N.A.		N.A.
Mean (SD)		534.0 (48.9)	
Median (IqR)		612.3 (612.3 to 626.5)	
DEX-i			0.0124
Mean (SD)	1431.6 (430.7)	1759.0 (423.6)	
Median (IqR)	1178.9 (1178.9 to 1571.9)	1571.9 (1571.9 to 2357.9)	
Total cost			<0.0001
Mean (SD)	2749.6 (844.5)	4009.7 (1381.6)	
Median (IqR)	2678.3 (2678.3 to 2678.3)	3465.2 (3465.2 to 4251.1)	

SD: Standard deviation; IqR: interquartile range; OCT: optical coherence tomography; N.A.: not applicable; Anti-VEGF: vascular endothelial growth factor inhibitors; DEX-i: dexamethasone intravitreal implant. * Odds Ratio was adjusted by lens status.

**Table 3 ijerph-20-05462-t003:** Relationship between probability of achieving a given visual acuity gain and treatment-group. The previously treated group was used as the reference for in all variables. Statistical analysis was performed using the Cochran–Mantel–Haenszel Odds Ratio test.

Outcome	Odds Ratio *	95% CI	*p*
BCVA gain ≥ 15 letters ** at month 12	3.55	1.09 to 11.58	0.0309
BCVA gain ≥ 15 letters at any time point measured	2.82	0.90 to 8.84	0.0658
BCVA gain ≥ 10 letters at month 12	2.63	0.82 to 8.45	0.1011
BCVA gain ≥ 10 letters at any time point measured	1.81	0.59 to 5.61	0.2903
BCVA gain ≥ 5 letters at month 12	2.59	0.50 to 13.40	0.2432
BCVA gain ≥ 5 letters at any time point measured	1.91	0.36 to 10.22	0.4369

* Odds Ratio was adjusted by lens status. ** Letters in the Early Treatment Diabetic Retinopathy Study (ETDRS) charts. BCVA: Best-corrected visual acuity, CI: confidence interval.

**Table 4 ijerph-20-05462-t004:** Cost-effectiveness ratio (CER) and incremental cost-effectiveness ratio (ICER) analysis considering the total direct costs according to the Andalusian Public Health System.

	Total Cost						
	Naïve *			Previously Treated *			
	Cost **	Outcome (Numerical)	CER	Cost **	Outcome (Numerical)	CER	ICER
BCVA gain ≥ 15 letters at any	2678.3	0.607	4411.86	3465.2	0.333	10,406.01	−5994.15
BCVA gain ≥ 15 letters at month 12	2678.3	0.607	4411.86	3465.2	0.286	12,116.08	−7704.22
BCVA gain ≥ 10 letters at any	2678.3	0.679	3944.48	3465.2	0.524	6612.98	−2668.50
BCVA gain ≥ 10 letters at month 12	2678.3	0.714	3751.12	3465.2	0.476	7279.83	−3528.71
BCVA gain ≥ 5 letters at any	2678.3	0.929	2882.99	3465.2	0.857	4043.41	−1160.04
BCVA gain ≥ 5 letters at month 12	2678.3	0.929	2882.99	3465.2	0.810	4278.03	−1395.04
Per letter gained ^†^	2678.3	17.5	153,05	3465.2	5.0	693.04	−539.99

* The median total cost was used in the analysis. ** Euros. ^†^ According to the median.

**Table 5 ijerph-20-05462-t005:** Adjusted mean change in best-corrected visual acuity (BCVA) and central retinal thickness (CRT) in treatment-naïve and previously treated eyes. The model included “Group” (Treatment-naïve versus previously treated) as a factor and age and length of DME as covariates.

	BCVA *			
	Treatment Naïve	Previously Treated	Difference	*p* ^a^
MCBCVAM2				0.0134
Mean (SE)	17.9 (2.5)	7.6 (2.9)	10.3 (4.0)	
95% CI	12.9 to 22.8	1.8 to 13.4	2.2 to 18.3	
MCBCVAM4				0.0517
Mean (SE)	10.9 (2.6)	2.4 (3.1)	8.5 (4.2)	
95% CI	5.6 to 16.1	−3.7 to 8.6	−0.06 to 16.9	
MCBCVAM6				0.4843
Mean (SE)	8.7 (3.1)	5.2 (3.6)	3.5 (5.0)	
95% CI	2.4 to 15.0	−2.2 to 12.5	−6.6 to 13.6	
MCBCVAM8				0.0060
Mean (SE)	17.9 (3.0)	4.1 (3.5)	13.8 (4.8)	
95% CI	11.9 to 23.8	−2.9 to 11.0	4.2 to 23.4	
MCBCVAM12				0.0092
Mean (SE)	20.4 (2.7)	8.7 (3.1)	11.7 (4.3)	
95% CI	15.1 to 25.8	2.5 to 15.0	3.0 to 20.3	
	**CRT ****			
	**Treatment Naïve**	**Previously Treated**	**Difference**	** *p* ** ** ^a^ **
MCCRTM2				0.5262
Mean (SE)	−193.5 (11.0)	−204.9 (12.9)	11.4 (17.8)	
95% CI	−215.8 to −171.3	−231.0 to −178.8	−24.5 to 47.3	
MCCRTM4				0.8123
Mean (SE)	−117.5 (22.4)	−126.0 (25.7)	8.5 (36.7)	
95% CI	−162.6 to −72.3	−177.9 to −74.1	−63.5 to 80.6	
MCCRTM6				0.0463
Mean (SE)	−131.1 (19.3)	−195.1 (22.7)	64.0 (31.2)	
95% CI	−170.1 to −72.1	−240.7 to −140.4	1.1 to 126.9	
MCCRTM8				0.1099
Mean (SE)	−197.9 (19.1)	−148.2 (21.9)	−49.6 (30.4)	
95% CI	−236.4 to −159.3	−192.5 to −104.0	−110.9 to 11.7	
MCCRTM12				0.0694
Mean (SE)	−211.4 (10.1)	−241.5 (11.8)	30.1 (16.2)	
95% CI	−231.7 to −191.2	−265.3 to −217.8	−2.5 to 62.7	

^a^ Analysis of covariance ANCOVA. * Additionally, adjusted by baseline BCVA and lens status. ** Additionally, adjusted by baseline CRT. MC: Mean change; BCVA: best-corrected visual acuity; M: month; SE: standard error; CI: confidence interval; CRT: central retinal thickness.

## Data Availability

The data that support the findings of this study are available from the corresponding author (M.M.-B.) upon reasonable request.

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
