# Peer review of "Cost-Effectiveness of Dexamethasone Intravitreal Implant in Naïve and Previously Treated Patients with Diabetic Macular Edema"

_ijerph, 2023, doi:10.3390/ijerph20085462_

Round 1

Reviewer 1 Report

Marta Medina-Baena et.al evaluated the cost-effectiveness of treating diabetic macular edema with dexamethasone intravitreal implant (DEX-i) in two distinct patient populations: naïve or previously treated with vascular endothelial growth factors inhibitors. Although the number of included patient sizes is limited, the study is generally well-structured and well-described. And it has the potential to guide adjusting therapeutic strategies for diabetic macular edema. I would like to suggest publication after addressing one issue described below.

It seems that the age of the patients in the naïve-treatment group is much younger than the previously treated group. Is age being adjusted in the analysis as a co-variant? If so, it is suggested to be described in the manuscript. If not, it is suggested to be discussed in the discussion section.

Author Response

We highly appreciate the reviewer comment. 

Here some relevant points: 
1. In methodology section, it is already specified that the difference in
BCVA and CRT have been adjusted:

"The analysis of covariance (ANCOVA) was  
used in the evaluation of the changes in BCVA and CRT between the  
study groups. The model included "Group" (Treatment-naïve versus  
previously treated) as a factor and age, length of DME, and lens  
status as covariates".

I would appreciate you indicate if there is something else we must add.

Reviewer 2 Report

In the present paper, Medina-Baena M. and co-workers provide a detailed cost effectiveness analysis of Dex-i in naive vs previously treated patients with DME, concluding that Dex-i proved to be more cost-effective in untreated eyes, thus suggesting that it should be taken into account as a first line therapy in some patients instead of anti-VEGF. For this reason, it should be interesting to perform cost analyses subdividing patients based on the presence/absence of SRF, as DME with subfoveal neuroretinal detachment is known to be a distinct "more inflammatory" pattern of DME, thus more prone to achieving better response after treatment with intravitreal steroids. Also discussion may be improved adding hypotheses on which group of patients with DME would benefit more from the use of Dex as first-line therapy.  

Unfortunately, as correctly stated by the Authors, the retrospective nature of the paper significantly limits its strength.

Reviewer 3 Report

There are patient costs expressed in the risks to which he is exposed by DEX-i implants; cataract or cortisone glaucoma.

Author Response

We highly appreciate the reviewer comment.

The authors have calculated the costs in the patients analyzed and during the first year of treatment. No cost estimate was made based on the risk of developing glaucoma or cataracts, if this was not the case for the patients at this fllow-up. If it is considered important to add this analysis, perhaps it could be done if the authors had more time to do so.

Round 2

Reviewer 2 Report

As suggested, the Authors carried out new analyses based on the presence/absence of SRF. They did not add new results in the manuscript, as they need more time. I suggest to improve methods and results with new data and to discuss them even if no relevant differences between the two groups emerged. As previously suggested, Authors should discuss on possible hypotheses on which group of patients with DME would benefit more from the use of Dex as first-line therapy, as different patterns of DME exist and inflammation is the main target of treatment with steroids. 

Author Response

Dear reviewer 2, The authors appreciate your suggestions and we are sure that the publication will improve its quality thanks to them. We have included in the article the results regarding the adjusted cost according to the presence or absence of SRF, despite there being no significant difference. In addition, in the discussion we have clarified the limitation of our study regarding this aspect, since it was not designed to analyze these aspects. The authors hope to have improved the quality of the article with this. Thank you.